inorganic chemistry/medicinal chemistry/physical chemistry

$[^{99m}Tc][Tc(CO)_3(H_2O)_3]^+$, HBED-CC, structural prediction, density functional theory

**Author for correspondence:**
De-Cai Fang
e-mail: dcfang@bnu.edu.cn

This article has been edited by the Royal Society of Chemistry, including the commissioning, peer review process and editorial aspects up to the point of acceptance.

# Synthesis of novel technetium-99m tricarbonyl-HBED-CC complexes and structural prediction in solution by density functional theory calculation

Shengyu Shi[1], Lifeng Yao[1,2], Linlin Li[1], Zehui Wu[3],
Zhihao Zha[4], Hank F. Kung[3,4], Lin Zhu[1]
and De-Cai Fang[1]

[1]College of Chemistry, Beijing Normal University, Beijing 100875, People's Republic of China
[2]College of Chemistry and Chemical Engineering, Qujing Normal University, Qujing 655011, People's Republic of China
[3]Beijing Institute of Brain Disorders, Capital Medical University, Beijing 100069, People's Republic of China
[4]Department of Radiology, University of Pennsylvania, Philadelphia, PA 19104, USA

D-CF, 0000-0003-3922-7221

HBED-CC (*N,N′*-bis[2-hydroxy-5-(carboxyethyl)benzyl]ethylene diamine-*N,N′*-diacetic acid, **L₁**) is a common bifunctional chelating agent in preparation of $^{68}Ga$-radiopharmaceuticals. Due to its high stability constant for the $Ga^{3+}$ complex ($logK_{GaL} = 38.5$) and its acyclic structure, it is well known for a rapid and efficient radiolabelling at ambient temperature with Gallium-68 and its high *in vivo* stability. $[^{99m}Tc][Tc(CO)_3(H_2O)_3]^+$ is an excellent precursor for radiolabelling of biomolecules. The aim of this study was to develop a novel preparation method of $^{99m}Tc$-HBED-CC complexes. In this study, HBED-CC-NI (2,2′-(ethane-1,2-diylbis((2-hydroxy-5-(3-((2-(2-nitro-1H-imidazol-1-yl)ethyl)amino)-3-oxopropyl)benzyl)-azanediyl))-diacetic acid, **L₂**), a derivative of HBED-CC, was designed and synthesized. Both **L₁** and **L₂** were radiolabelled by $[^{99m}Tc][Tc(CO)_3(H_2O)_3]^+$ successfully for the first time. In order to explore the coordination mode of metal and chelates, non-radioactive $Re(CO)_3\mathbf{L_1}$ and $Re(CO)_3\mathbf{L_2}$ were synthesized and characterized spectroscopically.

Tc(CO)$_3$L$_1$ and Tc(CO)$_3$L$_2$ in solution were calculated by density functional theory and were analysed with radio-HPLC chromatograms. It showed that [$^{99m}$Tc]Tc(CO)$_3$L$_2$ forms two stable diastereomers in solution, which is similar to those of [$^{68}$Ga]Ga-HBED-CC complexes. Natural bond orbital analysis through the natural population charges revealed a charge transfer between [$^{99m}$Tc][Tc(CO)$_3$]$^+$ and L$_1$ or L$_2$. The experimental results showed that tricarbonyl technetium might form stable complex with HBED-CC derivatives, which is useful for the future application of using HBED-CC as a bifunctional chelating agent in developing new $^{99m}$Tc-radiopharmaceuticals as diagnostic imaging agents.

## 1. Introduction

Positron emission tomography (PET) and single photon emission computed tomography (SPECT) are the two main nuclear medicine imaging techniques, which detect the metabolism of biomolecules, receptors and neuronal function in living tissues or organs. They have been widely used in the diagnosis of various diseases and assisting new drug development [1,2]. Technetium-99m with its reasonable physical half-life and optimal medium gamma-ray energy ($t_{1/2} = 6$ h, $\gamma = 140$ keV) is currently the most widely used isotope for SPECT imaging.

Chelators used for radiometal-based radiopharmaceuticals are typically covalently linked to a biologically active targeting molecule for making an active radiopharmaceutical agent. The chelator tightly binds to a radiometal ion. When the resulting complexes are injected into patients, the targeting molecule can deliver the isotope to different organs via blood circulation, effectively distributing site-specific radioactive probes for *in vivo* imaging. Currently, the most widely used PET radiopharmaceutical is 2-[$^{18}$F]fluoro-deoxy-glucose ([$^{18}$F]FDG), and it is the most commonly employed PET imaging agent for the diagnosis and monitoring the progression of cancer, in which the tumour tissue is using an abundant amount of glucose for excessive growth. However, in the past 10 years, with the commercialization of the $^{68}$Ge/$^{68}$Ga generator, the positron-emitting radionuclide, Gallium-68 ($t_{1/2} = 68$ min, $\beta^+$, 511 keV), has become a common isotope for new PET radiopharmaceutical research and development which can be produced without a nearby cyclotron [3,4]. In 2016, [$^{68}$Ga]Ga-DOTA-TATE [5] (Netspot) was approved by the FDA for the diagnosis of neuroendocrine tumour, an event that leads to a widespread clinical application and heightens the importance of this type of metal complexes in various clinical applications. Strong chelators, which bind to Ga(III), are commonly used in the preparation of $^{68}$Ga-radiopharmaceuticals. They include DOTA (1,4,7,10-tetraazacyclododecane-1,4,7,10-tetraacetic acid, $\log K_{GaL} = 21.3$) [6–8], NOTA (1,4,7-triazacyclononane-1,4,7-triacetic acid, $\log K_{GaL} = 31.0$) [9–11] and HBED-CC, etc. The latter chelator, HBED-CC, is an acyclic chelator based on an EDTA-type structure with two additional phenol coordinating covalent bonds for chelating Ga (III) (figure 1). The acyclic structure, HBED-CC, is well known for the rapid and efficient formation of stable metal (III) complexes at ambient temperature. The resulting HBED-CC Ga(III) complex displays high *in vivo* stability constant, which was reported as $\log K_{GaL} = 38.5$ [12–14]. In recent years, there have been numerous reports on [$^{68}$Ga]Ga-PSMA-11, which is an HBED-CC-based imaging agent targeting prostate-specific membrane antigen (PSMA) expression in prostate cancer [15–17]. Other reports also suggested usefulness of many other HBED-CC-based PET imaging agents [3,18–21].

Normally, $^{99m}$Tc-radiopharmaceuticals are often prepared by combining reactions of reducing agents, suitable complexing ligands and [$^{99m}$Tc]pertechnetate, which was obtained by eluting a $^{99}$Mo/$^{99m}$Tc generator with normal saline. It is a prerequisite to use different reducing agents and complexing chelates simultaneously to reduce technetium (VII) to lower valence states prior to the complex formation. Common technetium cores for different valence states are: technetium (V) such as (TcO)$^{3+}$ [22], (TcN)$^{2+}$ [23], (TcO$_2$)$^+$ [24], (Tc-HYNIC)$^{2+}$ [25], technetium (I), [$^{99m}$Tc]Tc(I)MIBI [26] and [$^{99m}$Tc][Tc(I)(CO)$_3$(H$_2$O)$_3$]$^+$. Except for [$^{99m}$Tc][Tc(CO)$_3$(H$_2$O)$_3$]$^+$, other technetium cores often require a complexing agent to provide four to five coordinate covalent bonds to form a stable complex. Although HBED-CC has enough coordination atoms to participate in the coordination, according to the calculation results of density functional theory (DFT), the configuration of HBED-CC caused a large steric hindrance resulting in unstable structures. In addition, the most commonly used reducing agent for forming these technetium cores is stannous salt, but former studies in our group have found that tin ion will compete with technetium in coordination with HBED-CC. Technetium tricarbonyl core might form a stable complex with only two to three coordination sites with HBED-CC, and the reducing agent is NaBH$_4$ [27].

In the past two decades, technetium tricarbonyl, [$^{99m}$Tc][Tc(CO)$_3$(H$_2$O)$_3$]$^+$, which only need two to three coordination sites to form a stable complex, has attracted great attention since it was first

**Figure 1.** The chemical structures of EDTA, HBED-CC, DOTA and NODAGA.

introduced as a precursor for radiolabelling of biomolecules by Alberto *et al.* [28]. There are several advantages of using [$^{99m}$Tc]Tc(CO)$_3$ to radiolabel biomolecules—easy preparation [29], easy substitution of its ligands in water media [30], smaller size [31] and inertness [32]. Considerable progress in recent years on [$^{99m}$Tc]Tc(CO)$_3$-radiolabelled bio-targeting molecular probes have been made, for example, technetium tricarbonyl radiolabelled NODAGA (1,4,7-triazacyclononane,1-glutaric acid-4,7-acetic) (figure 1) somatostatin receptor-targeting bioconjugate, in which NODAGA is also a popular ligand for Gallium-68 [33–36]. To our knowledge, HBED-CC complexes of [$^{99m}$Tc]Tc(CO)$_3$ have not been reported previously. Based on the current research results of HBED-CC-based Gallium-68 PET imaging agents, it would be interesting to use this readily available chelating ligand [18–21] to develop novel $^{99m}$Tc-radiopharmaceuticals. Accordingly, we have investigated the feasibility of using HBED-CC ligands for complexing with [$^{99m}$Tc][Tc(CO)$_3$(H$_2$O)$_3$]$^+$.

Due to the fact that $^{99}$Tc is a radioactive metal, there is no easy approach to establish the chemical structures of chelation between [$^{99m}$Tc][Tc(CO)$_3$(H$_2$O)$_3$]$^+$ **L$_1$** and **L$_2$** (figure 2) in solution. Since [$^{99m}$Tc]Tc(CO)$_3$ HBED-CC complexes are prepared in a high specific activity state and the chemical amount is in sub-nanomole quantity, which is amenable for regular chemical characterization. The chemical characterization of [$^{99m}$Tc][Tc(CO)$_3$(H$_2$O)$_3$]$^+$ **L$_1$** and **L$_2$** was performed with [Re(CO)$_3$(H$_2$O)$_3$]$^+$ **L$_1$** and **L$_2$** instead (Re is a non-radioactive surrogate of technetium-99m) [33]. However, structures of **L$_1$** and **L$_2$** have multiple flexible carboxylic acid chains and it is difficult to obtain the crystal structures of Re(CO)$_3$**L$_1$**/**L$_2$**. The possible coordination structures of [$^{99m}$Tc]Tc(CO)$_3$**L$_1$**/**L$_2$** were first determined by computational methods using DFT.

In recent years, many examples have been reported on using DFT, natural bond orbital (NBO) theory [37] in the evaluation and analysis of radiometal radiolabelled radiopharmaceuticals. It provides a theoretical method to calculate the interaction energies of the complexation of the metal core and the ligand [38–40]. This method might play a significant role in guiding the structural optimization of radiopharmaceuticals. Skelton *et al.* reported a series of theoretical studies on the formation of metal-NOTA complexes of Cu$^{2+}$, Ga$^{3+}$, Sc$^{3+}$ and In$^{3+}$. The chelation interaction energies in solution, short-range interatomic distances, NBO, chemical shifts and DFT-based reactivity-related properties revealed a significant level of interaction between NOTA and radiometal ions [41]. In addition, our research group has also evaluated related transition state of technetium-99m radiolabelled compounds in radiolabelling process by DFT previously, so it was sufficient theoretical basis in applying this method for studying [$^{99m}$Tc]Tc(CO)$_3$**L$_1$**/**L$_2$** [42].

Reported herein are the results of evaluation of chelation between [$^{99m}$Tc][Tc(CO)$_3$(H$_2$O)$_3$]$^+$ **L$_1$** and **L$_2$**, optimization of radiosynthesis parameters to obtain radiolabelled products with high radiochemical purity and the theoretical calculation using DFT for predicting the chemical structures in solution, and the results will be compared with radio-HPLC chromatograms of [$^{99m}$Tc]Tc(CO)$_3$**L$_1$**/**L$_2$** to predict the isomer formation.

**Figure 2.** The chemical structures of **L₁** and **L₂**.

# 2. Material and methods

## 2.1. Materials

Chemicals were purchased from commercial sources and used without further purification. Analytical grade solvents were used without further purification, unless otherwise specified. $^{99}Mo/^{99m}Tc$ generator was obtained from the China Institute of Atomic Energy (CIAE, Beijing, China). HPLC analysis was performed on a Shimadzu LC-20AT 230 V absorbance dual $\lambda$ detector (Milford, MA, USA) with a reversed-phase column Phenomenex luna C18 $250 \times 4.6$ mm. HRMS was performed on AB Sciex 5600 Plus. $^{1}H$ NMR spectra were recorded on a Bruker Avance spectrometer at 400 or 600 MHz. IR spectra were recorded as KBr pellets on an IRAffinity-1 Fourier transform infrared spectrometer in the region of 4000–500 cm$^{-1}$. Radioactivity was counted on a Cobra II autogamma counter (Perkin Elmer).

## 2.2. Computational methods

All the calculations in this study were performed under the Gaussian 09 program package [43]. Geometry structures of reactants were optimized using the Becke-3-Lee-Yang-Parr [44], in conjunction with the DGDZVP basis set [45]. All the optimized stationary points have been identified as minima (zero imaginary frequencies) via the vibrational analyses at the same level. The solvent (water) effect was considered by the polarized continuum model (PCM) [46] using our defined IDSCRF radii [47,48]. The Gibbs free energies were corrected by our THERMO program [49] at 368.15 K. The single-point enthalpies and energies calculations with dispersion interaction effects were proposed as B3LYP-D3 [50,51] with SMD [52] solvent model shown in support information.

## 2.3. Synthesis of chelators and Re-complexes

The synthesis of **L₁**, **L₂** and Re-complexes were shown in figure 3. The synthetic route from **a** to **e** was similar to previous methods reported [53].

Di-*tert*-butyl 2,2′-(ethane-1,2-diylbis((2-hydroxy-5-(3-((2-(2-nitro-1H-imidazol-1-yl)ethyl)amino)-3-oxo-propyl)benzyl)za-anediyl))diacetate (**8**). To a solution of **7** (433.8 mg, 0.673 mmol) in 10 ml DMF was added *N,N*-diisopropylethylamine (DIPEA, 1.04 g, 8 mmol), 1-hydroxybenzotriazole hydrate (HOBt, 270 mg, 2 mmol), *N*-(3-dimethylaminopropyl)-*N*-ethylcarbodiimide hydrochloride (EDC, 381 mg, 2 mmol) and 2-(2-nitro-1H-imidazol-1-yl)ethan-1-amine(235 mg, 1.345 mmol) at 0°C. The mixture was

Reagent and conditions: (a) MeOH, BF₃·Et₂O, rt; (b) (CHO)n, MgCl₂, Et₃N, ACN, reflux; (c) ethylenediamine, NaBH₄, MeOH, 50°C, rt; (d) *tert*-Butyl bromoacetate, Na₂CO₃, ACN, 60°C; (e) NaOH, MeOH, H₂O, rt; (f) DIPEA, HOBt, EDC, DMF, rt; (g) TFA, rt; (h)TFA, rt; (i) Re(CO)₃(H₂O)₃Br, 95°C.

**Figure 3.** The synthetic route of $L_1$, $L_2$, Re(CO)₃$L_1$ and Re(CO)₃$L_2$.

stirred at room temperature (rt) for 4 h. EtOAc (30 ml) was added to the reaction mixture. The mixture was then washed with $H_2O$ (10 ml × 2) as well as brine (10 ml), dried over $MgSO_4$ and filtered. The filtrate was concentrated and the residue was purified by FC (DCM/MeOH/NH₄OH = 95/5/0.5) to give 375 mg colourless oil **8** (yield: 60.6%): ¹H NMR (400 MHz, D₂O) $\delta$: 9.54 (s, 2H), 7.02–7.00 (m, 2H), 6.80 (s, 2H), 6.71 (t, 6H, $J = 5.6$), 6.58 (s, 2H), 4.49 (t, 4H, $J = 5.6$), 3.64–3.60 (m, 8H), 3.21 (s, 4H), 2.85 (t, 4H, $J = 7.2$), 2.66 (s, 4H), 2.48 (t, 4H, $J = 7.0$), 1.47 (s, 18H). LRMS calcd. For $C_{44}H_{60}N_{10}O_{12}$ $(M + H)^+$: 920.4, found 920.4.

2,2′-(ethane-1,2-diylbis((2-hydroxy-5-(3-((2-(2-nitro-1H-imidazol-1-yl)ethyl)amino)-3-oxopropyl) benzyl)-azanediyl))-diacetic acid (**2($L_2$)**). A solution of **8** (375 mg, 0.408 mmol) in 10 ml TFA was stirred at rt for 4 h. The reaction mixture was evaporated in vacuum and the residue was purified by prep-HPLC to give 197 mg white solid **2($L_2$)** (yield: 59.8%): ¹H NMR (600 MHz, D6-DMSO) $\delta$: 7.32 (s, 2H), 7.06 (s, 2H), 6.89 (s, 2H), 6.87 (t, 2H, $J = 3$), 6.64 (d, 2H, $J = 6$), 4.38 (t, 4H, $J = 6$), 3.72 (s, 4H), 3.43–3.40 (m, 4H), 3.35 (t, 4H, $J = 6$), 3.26 (s, 4H), 2.57 (t, 4H, $J = 6$), 2.21 (t, 4H, $J = 9$). LRMS calcd. For $C_{36}H_{44}N_{10}O_{12}$ $(M + H)^+$: 809.3, found 809.3.

Synthesis of Re(CO)₃$L_1$/$L_2$. A solution of pentacarbonylrhenium bromide (80 mg, 0.2 mmol) in 10 ml water was refluxed at 105°C for 24 h. $L_1$/$L_2$ (20/30 mg, 0.038 mmol) was then added to above reaction solution (2 ml) containing Re(CO)₃(H₂O)₃Br intermediate. The mixture was acidified with 0.1 N HCl till pH = 5 and heated at 95°C for 3 h. The reaction solution was loaded onto an Oasis HLB SPE cartridge (6 cc, 150 mg, conditioned with 10 ml of ethanol followed by 10 ml of $H_2O$), which was then washed with 30 ml $H_2O$. The product was eluted with 2 ml methanol and the solvent was evaporated to dryness. The remaining solid was washed with acetonitrile (2 × 2 ml) and a white powder was obtained (yield: 23.1%) Re(CO)₃$L_2$: ¹H NMR (400 MHz, D6-DMSO) $\delta$: 7.91 (d, 2H, $J = 8$), 7.30 (d, 2H, $J = 8$), 7.08 (t, 4H, $J = 4$), 6.79 (d, 2H, $J = 8$), 6.62 (d, 2H, $J = 8$), 4.40 (t, 4H, $J = 6$), 4.00 (s, 2H), 3.41 (s, 4H), 2.63–2.57 (m, 10H), 2.29–2.17 (m, 8H). HRMS calcd. For $ReC_{39}H_{43}N_{10}O_{15}$ $(M + H)^+$: 1079.2, found 1079.2792.

## 2.4. Preparation of $[^{99m}Tc][Tc(CO)_3(H_2O)_3]^+$ precursor

$Na_2CO_3$ (4.0 mg), $NaBH_4$ (5.6 mg) and potassium sodium tartrate (20.0 mg) were dissolved in 1 ml saline and the solution was purged with CO for 15 min. $[^{99m}Tc]NaTcO_4$ (370 MBq) in 1 ml saline was then added and the mixture was stirred at 80°C for 30 min. After cooling to room temperature, the solution was acidified with 0.05 mol l⁻¹ HCl until pH = 7–8. The RCP was determined using radio-HPLC and radio-TLC. The radio-TLC was performed on a polyamide strip and developed with acetonitrile. Radio-TLC results showed $[^{99m}Tc][Tc(CO)_3(H_2O)_3]^+$ exhibited Rf = 0.1–0.2, while the $[^{99m}Tc]TcO_4^-$ displayed Rf = 0.3–0.6.

## 2.5. Radiosynthesis and reaction parameters optimization of [$^{99m}$Tc]Tc(CO)$_3$**L$_1$/L$_2$**

Optimal reaction parameters were determined through various temperatures (20–100°C), pH values (4–10) and at a ligand concentration ranging from 0.05 to 1.00 mM.

### 2.5.1. pH effect

To a solution of [$^{99m}$Tc][Tc(CO)$_3$(H$_2$O)$_3$]$^+$ (37 MBq) in 1 ml saline was added **L$_1$/L$_2$** (0.50/0.76 mg, 0.94 µmol) in 1 ml 0.05 M NaOAc. The final pH value is 5.0. Other pH value was obtained with 0.5 M AcOH or 0.5 M NaOH. The mixture was heated at 95°C for 30 min. The RCP was determined by radio-HPLC. [$^{99m}$Tc]Tc(CO)$_3$**L$_1$/L$_2$** were eluted applying different gradients of 0.05 M (v/v) TEAP in H$_2$O and methanol at a constant flow of 1 ml min$^{-1}$ (0–10 min, from 90% H$_2$O with 0.05 M TEAP to 100% methanol and then back to 90% H$_2$O with 0.05 M TEAP 15–20 min). The analytical methods of the following experiments are the same.

### 2.5.2. Precursor concentration effect

To a solution of [$^{99m}$Tc][Tc(CO)$_3$(H$_2$O)$_3$]$^+$ (37 MBq) in 1 ml saline was added **L$_1$/L$_2$** [0.05/0.076 mg (0.09 µmol), 0.10/0.15 mg (0.18 µmol), 0.25/0.38 mg (0.47 µmol), 0.50/0.76 mg (0.94 µmol) and 1.00/1.52 mg (1.9 µmol) in 1 ml 0.05 M NaOAc, pH = 5]. The reaction mixture was heated at 95°C for 30 min.

### 2.5.3. Temperature effect

To a solution of [$^{99m}$Tc][Tc(CO)$_3$(H$_2$O)$_3$]$^+$ (37 MBq) in 1 ml saline was added to **L$_1$/L$_2$** (0.50/0.76 mg, 0.94 µmol) in 1 ml 0.05 M NaOAc, pH = 5. The reaction mixture was heated at 20, 40, 60, 80, 95 and 100°C for 30 min.

## 2.6. *In vitro* stability in PBS and BSA of [$^{99m}$Tc]Tc(CO)$_3$**L$_2$**

In total, 740 kBq of [$^{99m}$Tc]Tc(CO)$_3$**L$_2$** was added to 1 ml phosphate-buffered saline (PBS, pH = 7.4, 0.1 M) or bovine serum albumin (BSA). The solutions were incubated at room temperature for PBS and 37°C for BSA. Then the sample was taken to radio-HPLC analysis to measure the stability in PBS and BSA at 2, 4 and 6 h.

## 2.7. Histidine and cysteine challenges of [$^{99m}$Tc]Tc(CO)$_3$**L$_2$**

In total, 740 kBq of [$^{99m}$Tc]Tc(CO)$_3$**L$_2$** was added to a solution of L-histidine or L-cysteine (500 µl, 1.0 mM) in PBS (10 mM, pH 7.4). The mixtures were incubated at 37°C for 6 h. Aliquots were analysed by radio-HPLC at 2, 4 and 6 h to determine the radiochemical purity.

## 2.8. Partition coefficient (log D$_{OW}$) of [$^{99m}$Tc]Tc(CO)$_3$**L$_2$**

One hundred microlitres of [$^{99m}$Tc]Tc(CO)$_3$**L$_2$** (7.4 MBq) was added to a mixture of PBS (pH = 7.4): *n*-octanol (1:1 v/v, 1.9 ml) and the mixture was shaken vigorously for 5 min at room temperature. The two layers were separated and the radioactivity associated with 100 µl aliquots of each layer was counted on a Cobra II autogamma counter (Perkin Elmer). The experiment was performed using three separate samples.

## 2.9. Electrophoresis experiments

PBS (pH = 7.4) was added to the electrophoresis tank. The voltage was set to 150 V. Xinhua No. 1 paper was cut to about 15 cm, placed in the electrophoresis tank and the strips immersed in PBS on both sides. The PBS gradually wet the paper strip, and the radiotracer liquid was spotted on the central position by a capillary tube when the front edge of the two liquids was about to coincide. It was placed for 15 min, and the paper strip removed. The point was taken as a centre, 5 mm to the left and right taken as centre. The counts of the positive electrode, the centre and the negative electrode, respectively, were measured.

# 3. Results and discussion

## 3.1. Synthesis of chelators and Re-complexes

Model chelator $L_2$ was synthesized by reacting 2-(2-nitro-1H-imidazol-1-yl)ethan-1-amine with $L_2(tBu)_2$, followed by deprotection to remove the *tert*-butyl groups. $L_1$ was synthesized directly by treating protected HBED-CC(tBu)$_2$ with TFA. $L_1$ and $L_2$ were characterized by LC-ESI-MS and $^1$H NMR. $Re(CO)_3L_1$ and $Re(CO)_3L_2$ were synthesized by reacting the $[Re(OH_2)_3(CO)_3]Br$ precursor with $L_1$ and $L_2$. HRMS showed the expected molecular ion mass, while the peak patterns matched the theoretical isotope distribution (shown in electronic supplementary material, figures S1 and S2). For $Re(CO)_3L_2$, two tight split peaks were found in UV-HPLC analysis (figure 4e). But for $Re(CO)_3L_1$, multiple peaks were found. The tricarbonyl rhenium displayed strong coordination ability with N and O atoms, while the unsubstituted HBED-CC showed a plurality of sites for coordination of tricarbonyl rhenium. Therefore, we conclude that there might be multiple binding isomers in $Re(CO)_3L_1$. Infrared spectra of $Re(CO)_3L_2$ revealed the typical very strong stretching bands of facially coordinated CO at 2021 cm$^{-1}$. It also showed strong absorptions at 1891 and 1638 cm$^{-1}$ attributed to the C=O stretch of the free carboxylate groups and nitroimidazole (shown in electronic supplementary material, figure S3). The figures of $^1$H NMR results are in electronic supplementary material, figures S4–S6.

## 3.2. Radiosynthesis and characterization of $[^{99m}Tc]Tc(CO)_3L_1/L_2$

$[^{99m}Tc][Tc(CO)_3(H_2O)_3]^+$ was prepared according to the publication of Alberto *et al.* [28]. Radiochemical purity (RCP) of $[^{99m}Tc][Tc(CO)_3(H_2O)_3]^+$ was more than 98% (figure 4b) and it was used in the following experiments directly. Comparisons of formation of $[^{99m}Tc]Tc(CO)_3L_1/L_2$ using different reaction conditions were performed (figure 5), between pH 4 and 10, at different reaction temperatures, between 20°C and 100°C, precursor concentration between 0.05 and 1.00 mM). It was found that the RCY for $[^{99m}Tc]Tc(CO)_3L_2$ was significantly reduced when the pH value was low (pH < 5). This might be due to insufficient ionization of the carboxylic acid under strong acidic conditions and the partial protonation of N atom, which inhibited the formation of coordination covalent bonds. In addition, under heating conditions, the stability of $[^{99m}Tc][Tc(CO)_3(H_2O)_3]^+$ precursor is poor, especially when pH > 7, which may also suppress the complexation reaction. The effect of the reaction temperature was measured using the optimal radiolabelling parameters described above (pH = 5, precursor concentration is 0.5 mM). Radiolabelling efficiency improved with increasing temperatures and pure product (RCP > 95%) can be obtained at 95–100°C. When $[^{99m}Tc]Tc(CO)_3L_2$ and $Re(CO)_3L_2$ were co-injected in HPLC, the radioactivity (13.9 min) and UV peak (13.8 min) (figure 4d,e) showed similar peak shape and retention time, which provided strong evidence for confirmation. A small delay in retention time on HPLC between UV and radioactivity signal was due to a sequential connection of the detectors. During the radiolabelling reaction of $[^{99m}Tc]Tc(CO)_3L_1$, the $[^{99m}Tc][Tc(CO)_3(H_2O)_3]^+$ intermediate has disappeared, but there are still four peaks in radio-HPLC chromatogram (Rt at 12.0 min, P4; 12.2 min, P3; 13.2 min, P1; 13.3 min, P2; figure 4c).

## 3.3. Structural analysis of $Tc(CO)_3L_1/L_2$ by DFT

The structure of $L_1$ and $L_2$ has multiple flexible carboxylic acid chains and it is difficult to obtain the crystal structures of $Re(CO)_3L_1/L_2$. In order to further confirm the coordination structures of the complexes in solution, in this study, DFT was used to calculate the interaction energies of the possible structures of $Tc(CO)_3L_1/L_2$.

For $[^{99m}Tc]Tc(CO)_3L_1$, four components were found in radio-HPLC chromatogram. Each retention time corresponds to a possible structure. We speculated that there are bonding isomers in the radiolabelling products. To further elucidate the possible structures, several possible structures of $Tc(CO)_3L_1$ were calculated by B3LYP + IDSCRF/DGDZVP method. According to the energies calculated by DFT, the chelation interaction energies ($\Delta G$) of the complex reaction were calculated using the following formula:

$$\Delta G = G(Tc(CO)_3L_1/L_2) + 3 \times G(H_2O)$$

$$- G(L_1/L_2) - G(Tc(CO)_3(H_2O)_3^+)$$

Among numbers of possible coordination modes of $Tc(CO)_3L_1$, five relatively stable structures were found and the corresponding chelation interaction energies are given in table 1. It can be realized from table 1 that the chelation interaction free energies ($\Delta G$) for these five structures **a1**, **a2**, **a3**, **a4** and **a5** are −14.4, −7.6, −4.3, −1.7 and −0.1 kcal mol$^{-1}$, respectively. Dispersion corrections (D3) did not change the

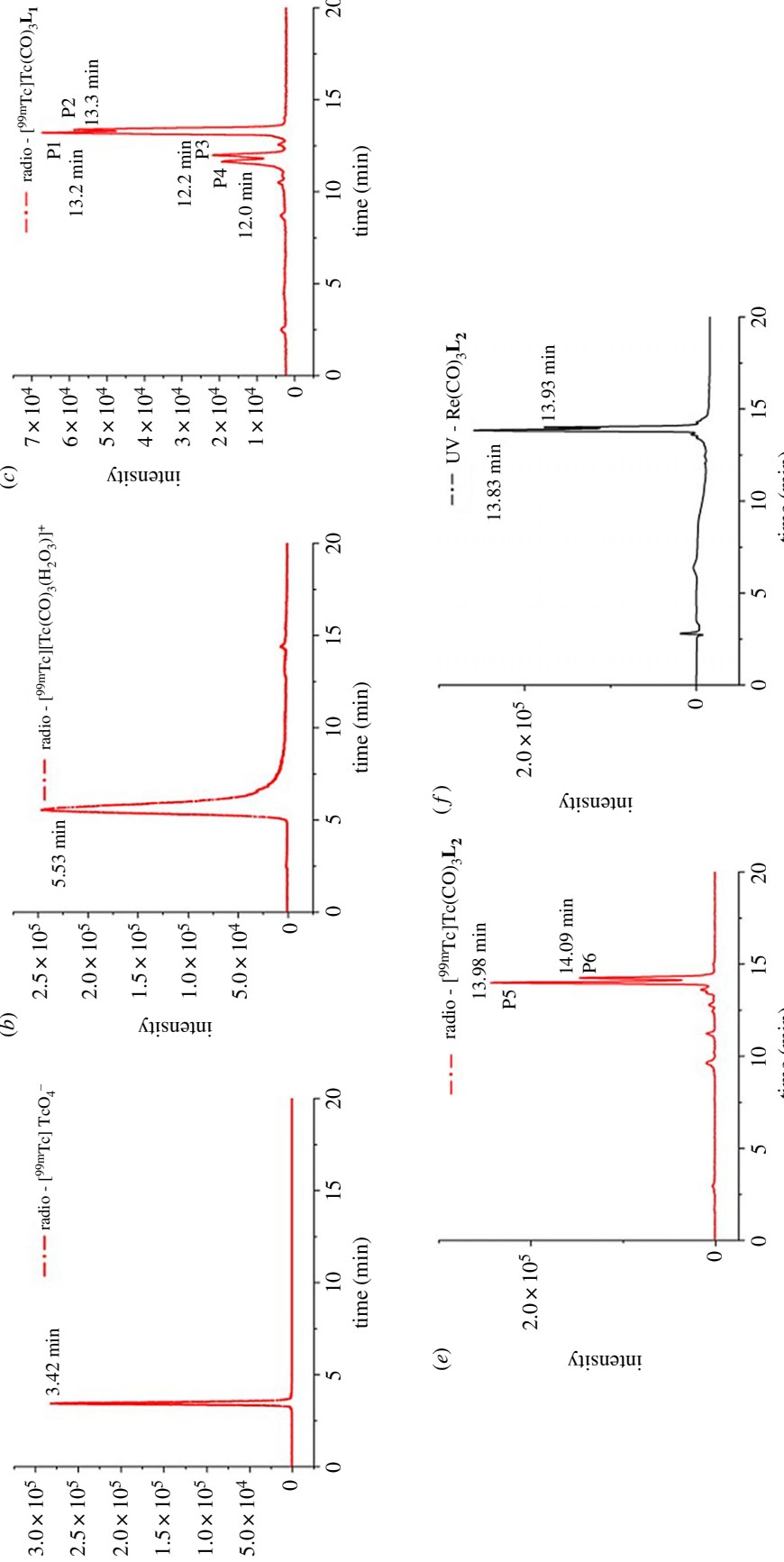

**Figure 4.** HPLC chromatogram of (a) [$^{99m}$Tc]TcO$_4^-$, (b) [$^{99m}$Tc][Tc(CO)$_3$(H$_2$O)$_3$]$^+$, (c) [$^{99m}$Tc]Tc(CO)$_3$**L$_1$**, (d) [$^{99m}$Tc]Tc(CO)$_3$**L$_2$** and (e) Re(CO)$_3$**L$_2$** (UV at 280 nm).

**Figure 5.** Radiosynthesis of [$^{99m}$Tc]Tc(CO)$_3$**L$_1$**/**L$_2$** (0.05–1.00 mM), the proposed coordination mode was calculated by DFT.

**Table 1.** Several possible coordination modes and chelation interaction energies calculated by B3LYP + IDSCRF/DGDZVP method at SMD solvent model for Tc(CO)$_3$**L$_1$** (denoted with **a**) and Tc(CO)$_3$**L$_2$** complexes (denoted with **b**) expressed in the form of $\Delta G$ (Gibbs free energies).

|  | coordination mode[a] | $\Delta G$ (kcal mol$^{-1}$) | $\Delta G$[b] (kcal mol$^{-1}$) |
|---|---|---|---|
| **a1** | N, N, O-4 | −14.4 | −22.9 |
| **a2** | N, N, O-6 | −7.6 | −19.8 |
| **a3** | O-4*, O7-1, O7-2 | −4.3 | −7.5 |
| **a4** | O-4, O7-1, O7-2 | −1.7 | 0.9 |
| **a5** | O7-1, O7-2 | −0.1 | 10.2 |
| **b1** | N, N, O-4 | −19.1 | −29.9 |
| **b2** | N, N, O-4 | −15.0 | −24.8 |
| **b3** | N, N, O-4 | −11.8 | −22.3 |

[a]Atoms that participate in the coordination on the ligand were shown, for example, O-4 refers to an O bonded to C-4 in figure 2.
[b]Dispersion corrected total Gibbs free energies with thermal correction calculated by B3LYP-D3 + IDSCRF/DGDZVP method at SMD solvent model.

order of the chelation interaction free energies for these five structures, and the obtained results are −22.9, −19.8, −7.5, 0.9 and 10.2 kcal mol$^{-1}$ (table 1), respectively. The order of the interaction energies for these structures is the same trend with those of interaction free energies, as stated in electronic supplementary material. Their corresponding structures are shown in figure 6, and the optimized coordinates of each structure were shown in electronic supplementary material. This can explain why [$^{99m}$Tc]Tc(CO)$_3$**L$_1$** has several peaks in radio-HPLC chromatogram. The highest peak P1 in figure 4c might come from isomer **a1** and other peaks might originate from isomers **a2**, **a3** and **a4**. This proves that the most stable coordination mode of Tc(CO)$_3$**L$_1$** in solution is N, N, O-4. This is consistent with the result in the published paper that N preferentially participates in the coordination of technetium-99m [36]. The peaks are split to P1 and P2; these two adjacent peaks may be two bonding isomers, but may also be caused by different conformations of the same coordination mode. The part about conformational isomers will be discussed in detail in structural analysis of Tc(CO)$_3$**L$_2$**.

For Tc(CO)$_3$**L$_2$**, since the carboxylic acid (C-7) at both ends of HBED-CC is protected, the complex does not have as many possible structures as Tc(CO)$_3$**L$_1$**. After DFT calculation, it was found that N, N, O-4 is the only stable coordination mode. However, there are three relatively stable conformations in this coordination mode (**b1**, **b2** and **b3**), which are shown in figure 6. The chelation interaction free energies ($\Delta G$) for these three conformations are −19.1, −15.0 and −11.8 kcal mol$^{-1}$ (without dispersion correction in table 1), and −29.9, −24.8 and −22.3 kcal mol$^{-1}$ (with dispersion correction in table 1). The results may explain why splitting peaks appear in the HPLC chromatogram of [$^{99m}$Tc]Tc(CO)$_3$**L$_2$**, which correspond to two different conformations, respectively. The highest peak P5 in figure 4d might

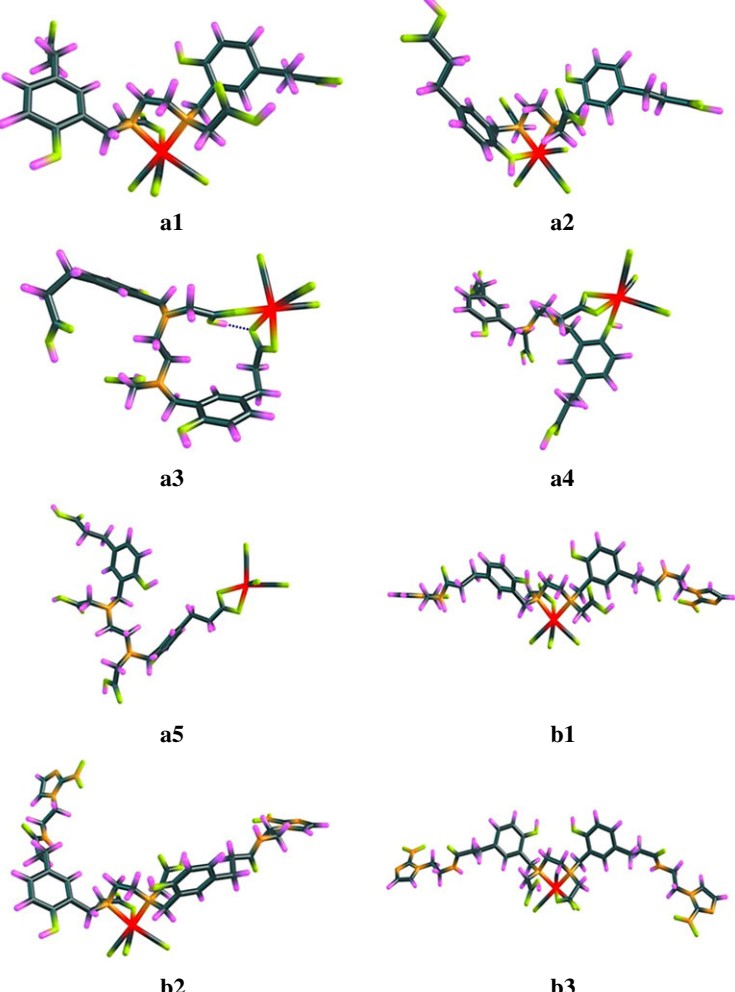

**Figure 6.** Possible coordination structures of Tc(CO)$_3$**L**$_1$ (**a1,a2,a3,a4,a5**) and Tc(CO)$_3$**L**$_2$ (**b1,b2,b3**) complexes characterized with DFT calculation.

comes from conformation **b1**, and P6 might originate from conformation **b2**. Diastereomers are very common in Gallium-68 radiolabelled HBED-CC complexes [54].

## 3.4. Analysis of frontier molecular orbital theory and natural bond orbital

The calculated HOMO–LUMO energies and $\Delta E_{LUMO-HOMO}$ gaps for the above four relatively stable structures of Tc(CO)$_3$**L**$_1$ were shown in table 2, from which one can realize that the decreasing order of HOMO–LUMO gaps for **a1**, **a2**, **a3** and **a4** have been found. Such order is in reverse proportion to the stabilities of **a1**, **a2**, **a3** and **a4**. NBO calculation will give the relatively accurate atomic charge and thus the charge transfer amount among the different fragments of the complexes was known in table 3, from which one can observe that the electron transfer direction during the coordination is from **L**$_1^-$ or **L**$_2^-$ to Tc(CO)$_3$ portion, i.e. the portion Tc(CO)$_3$ (UNIT 1) becomes less positive when the complex Tc(CO)$_3$**L**$_1$ or Tc(CO)$_3$**L**$_2$ is formed.

## 3.5. *In vitro* studies of [$^{99m}$Tc]Tc(CO)$_3$**L**$_2$

Using [$^{99m}$Tc]Tc(CO)$_3$**L**$_2$ as a model testing compound for complex formation, a preliminary study on the *in vitro* physical and chemical properties of [$^{99m}$Tc]Tc(CO)$_3$**L**$_2$ were evaluated. [$^{99m}$Tc]Tc(CO)$_3$**L**$_2$ displayed high stability in PBS (phosphate buffer saline) buffer (10 mM, pH = 7.4) at room temperature and BSA at 37°C through 6 h, as confirmed by radio-HPLC (figure 7). Challenge experiments against 1 mM L-histidine or L-cysteine at 37°C showed no transchelation and excellent stability (RCP > 95%, by radio-HPLC) for [$^{99m}$Tc]Tc(CO)$_3$**L**$_2$ through 6 h (figure 7). The *n*-octanol/water distribution coefficient

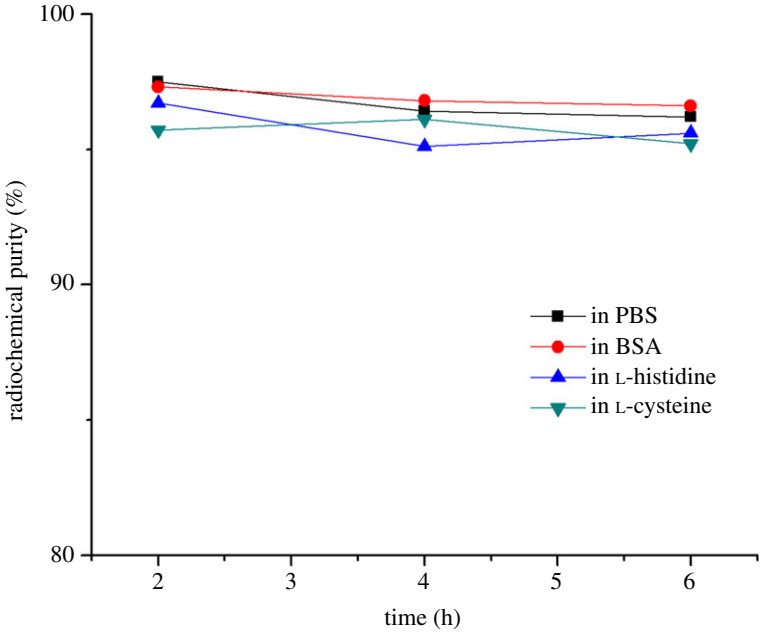

**Figure 7.** The stability in PBS, BSA, L-histidine and L-cysteine of $[^{99m}Tc]Tc(CO)_3L_2$. ($n = 2$).

**Table 2.** The energy of HOMO/LUMO molecular orbitals of four possible structures of $Tc(CO)_3L_1$.

| energy (eV) | a1 | a2 | a3 | a4 |
|---|---|---|---|---|
| HOMO | −6.359 | −6.046 | −5.820 | −5.624 |
| LUMO | −1.442 | −1.595 | −1.815 | −1.921 |
| $\Delta E_{LUMO-HOMO}$ | 4.917 | 4.452 | 4.005 | 3.703 |

**Table 3.** Sum of natural atomic charge of the portion $Tc(CO)_3^+$ (UNIT 1) and $L_1^-/L_2^-$ (UNIT 2) for **a1 a2 a3 a4** and **b1**.

| | | a1 | a2 | a3 | a4 | b1 | $L_1^-/L_2^-$ | $Tc(CO)_3^+$ |
|---|---|---|---|---|---|---|---|---|
| sum of NAC | UNIT 1 | 0.39 | 0.34 | 0.37 | 0.36 | 0.39 | — | 1[a] |
| | UNIT 2 | −0.39 | −0.34 | −0.37 | −0.36 | −0.39 | −1[b] | — |

[a]Before the coordination, the charge of $Tc(CO)_3$ is +1.
[b]Before the coordination, the charge of $L_1/L_2$ is −1.

(log $D_{OW}$) value of $[^{99m}Tc]Tc(CO)_3L_2$ determined at pH = 7.4 was found to be −1.73 ± 0.05 ($n = 3$), which revealed high hydrophilicity of complex. Electrophoresis experiments showed that $[^{99m}Tc]Tc(CO)_3L_2$ was electrically neutral, while the negatively charged $[^{99m}Tc]TcO_4^-$ showed a high radioactivity concentration in anode.

## 4. Conclusion

In this study, we reported the first example of the formation of technetium tricarbonyl complexes with HBED-CC($L_1$) and HBED-CC-NI($L_2$). After optimization of reaction parameters, including pH value, temperature and precursor concentration, $[^{99m}Tc]Tc(CO)_3L_1$ and $[^{99m}Tc]Tc(CO)_3L_2$ were accomplished (RCP > 95%) and showed high stability *in vitro*. Re(CO)$_3L_1/L_2$ were synthesized and characterized spectroscopically as standards to validate the radiolabelled compounds. A component analysis of the radio-HPLC chromatograms of the two radiolabelled compounds and the perspective of computational chemistry by DFT suggested that they are consistent. The theoretical calculation predicted that the peaks of the four components in the radio-HPLC chromatograms of $[^{99m}Tc]Tc(CO)_3L_1$ were probably due to the presence of

bonding isomers in solution. For $[^{99m}Tc]Tc(CO)_3L_2$, there was only one stable coordination mode, but two conformations with similar chelation interaction energies were found. This finding was consistent with peaks observed by HPLC chromatograms of $[^{99m}Tc]Tc(CO)_3L_2$ and $Re(CO)_3L_2$. Chelation interaction energies in solution, sum of natural atomic charge (NAC), HOMO/LUMO energies and $\Delta E_{LUMO-HOMO}$ gap revealed a significant level of interaction between tricarbonyl technetium core and $L_1/L_2$. In summary, the complex structure of $[^{99m}Tc]Tc(CO)_3L_1$ was evaluated and isomers in solution were probably formed. However, the most stable coordination mode N, N, O-4 calculated by FDT calculation was consistent with the conclusion in the published paper with nitrogen atoms preferentially participated in the coordination of technetium-99m tricarbonyl core. When two carboxylic acid arms at both ends of HBED-CC (C-7) are substituted (such as targeting groups), the complex has only one stable coordinated complex (N, N, O-4). For $[^{99m}Tc]Tc(CO)_3$-HBED-based complexes, coordination mode of N, N, O-4 was preferred. This approach might have great significance for the future development of $^{99m}Tc$-radiopharmaceuticals.

Data accessibility. The datasets supporting this article have been uploaded as part of the electronic supplementary material.

Authors' contributions. S.S., L.L., Z.W, Z.Z., L.Z. and H.F.K.: synthesis and radio labelling. L.Y. and D.-C.F.: DFT calculation.

Competing interests. We declare we have no competing interests.

Funding. This work was supported in part by grants from the National Key Research and Development Program of China (grant nos. 2016YFC1306304 and 2018YFC1312302), the Key Field Research and Development Program of Guangdong Province (grant no. 2018B030337001) and Beijing Natural Science Foundation of Key Project (grant no. 7171002).

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
