## [Reviewer comments · Royal Society Open Science]

Review History

RSOS-191247.R0 (Original submission)

Review form: Reviewer 1

Is the manuscript scientifically sound in its present form?

Yes

Are the interpretations and conclusions justified by the results?

Yes

Is the language acceptable?

Yes

Do you have any ethical concerns with this paper?

No

Have you any concerns about statistical analyses in this paper?

No

Recommendation?

Accept with minor revision (please list in comments)

Comments to the Author(s)

This manuscript outlined the synthesis of technetium tricarbonyl complexes with N,O-based ligands, combined with density functional calculations to analyze the interaction energies and frontier molecular orbitals. This is a very nice example of combined experimental and computational work. It may be published after addressing the following minor issues:

1. The standard B3LYP functional was used for the calculations. The only concern is that dispersion was not included, and single point calculations with Grimme's dispersion correction should be added.
2. It seems that electronic energies or enthalpies were reported, but Gibbs energies may be reported for comparison.
3. Solvation effect in water (if in aqueous solution) should be considered using the SMD solvation model.
4. Coordinates should be given in the supporting information.

Review form: Reviewer 2

Is the manuscript scientifically sound in its present form?

No

Are the interpretations and conclusions justified by the results?

Yes

Is the language acceptable?

Yes

Do you have any ethical concerns with this paper?

No

Have you any concerns about statistical analyses in this paper?

No

Recommendation?

Major revision is needed (please make suggestions in comments)

Comments to the Author(s)

The authors explained the synthesis and structural study of two complexes including technetium-99m with multidentate ligands.

It seems that there are many errors and incorrect descriptions in the manuscript. The authors should pay more attention to make manuscript.

I feel this manuscript is not suitable for Royal Society Open Science at present version.

I recommend acceptance of the paper if the authors manage to address some major issues.

For example, the IR spectrum in the electronic supplementary material is upside down !

In addition, in the HRMS spectrum (Fig2 in ESM) of $[\text{Re}(\text{CO})_3\text{L}_2]$, there is no data corresponding to the measured values described in the experimental section (HRMS calcd. For $\text{ReC}_3\text{9H}_4\text{3N}_1\text{0O}_1\text{5} (\text{M}+\text{H})^+$: 1079.7, found 1079.7827). Please check other experimental data described in the ESM.

For the DFT calculation results, the coordinates of the calculation results for each structure should be described in ESM. If authors had performed the calculations in the solution, the authors should refer to the details of how to handle calculations in solution (for example, the polarizable continuum model (PCM) to include solvent effects).

Anyway, the authors should pay attention to the the manuscript to make revised version.

Decision letter (RSOS-191247.R0)

14-Aug-2019

Dear Dr Fang:

Title: Synthesis of novel technetium-99m tricarbonyl-HBED-CC complexes and structural prediction in solution by density functional theory
Manuscript ID: RSOS-191247

The editor assigned to your manuscript has now received comments from reviewers. We would like you to revise your paper in accordance with the referee and Subject Editor suggestions which can be found below (not including confidential reports to the Editor). Please note this decision does not guarantee eventual acceptance.

Please submit your revised paper before 06-Sep-2019. Please note that the revision deadline will expire at 00.00am on this date. If we do not hear from you within this time then it will be assumed that the paper has been withdrawn. In exceptional circumstances, extensions may be possible if agreed with the Editorial Office in advance. We do not allow multiple rounds of revision so we urge you to make every effort to fully address all of the comments at this stage. If deemed necessary by the Editors, your manuscript will be sent back to one or more of the original reviewers for assessment. If the original reviewers are not available we may invite new reviewers.

Please also include the following statements alongside the other end statements. As we cannot

publish your manuscript without these end statements included, if you feel that a given heading is not relevant to your paper, please nevertheless include the heading and explicitly state that it is not relevant to your work.

- Acknowledgements

RSC Associate Editor:
Comments to the Author:
(There are no comments.)

RSC Subject Editor:
Comments to the Author:
(There are no comments.)

Reviewers' Comments to Author:
Reviewer: 1

Comments to the Author(s)

This manuscript outlined the synthesis of technetium tricarbonyl complexes with N,O-based ligands, combined with density functional calculations to analyze the interaction energies and frontier molecular orbitals. This is a very nice example of combined experimental and computational work. It may be published after addressing the following minor issues:

1. The standard B3LYP functional was used for the calculations. The only concern is that dispersion was not included, and single point calculations with Grimme's dispersion correction should be added.
2. It seems that electronic energies or enthalpies were reported, but Gibbs energies may be reported for comparison.
3. Solvation effect in water (if in aqueous solution) should be considered using the SMD solvation model.
4. Coordinates should be given in the supporting information.

Reviewer: 2

Comments to the Author(s)

The authors explained the synthesis and structural study of two complexes including technetium-99m with multidentate ligands.

It seems that there are many errors and incorrect descriptions in the manuscript. The authors should pay more attention to make manuscript.

I feel this manuscript is not suitable for Royal Society Open Science at present version.

I recommend acceptance of the paper if the authors manage to address some major issues.

For example, the IR spectrum in the electronic supplementary material is upside down !

In addition, in the HRMS spectrum (Fig2 in ESM) of $[\text{Re}(\text{CO})_3\text{L}_2]$, there is no data corresponding to the measured values described in the experimental section (HRMS calcd. For $\text{ReC}_39\text{H}_43\text{N}_{10}\text{O}_{15}(\text{M}+\text{H})^+$: 1079.7, found 1079.7827). Please check other experimental data described in the ESM.

For the DFT calculation results, the coordinates of the calculation results for each structure should be described in ESM. If authors had performed the calculations in the solution, the authors should refer to the details of how to handle calculations in solution (for example, the polarizable continuum model (PCM) to include solvent effects).

Anyway, the authors should pay attention to the the manuscript to make revised version.

Author's Response to Decision Letter for (RSOS-191247.R0)

See Appendix A.

RSOS-191247.R1 (Revision)

Review form: Reviewer 2

Is the manuscript scientifically sound in its present form?

Yes

Are the interpretations and conclusions justified by the results?

Yes

Is the language acceptable?

Yes

Do you have any ethical concerns with this paper?

No

Have you any concerns about statistical analyses in this paper?

No

Recommendation?

Accept as is

Comments to the Author(s)

Comments to the Author

I appreciate the effort of the revised manuscript presented following the suggestions from reviewers.

The manuscript has been revised well. I think this manuscript will be acceptable.

Decision letter (RSOS-191247.R1)

04-Oct-2019

Dear Dr Fang:

Title: Synthesis of novel technetium-99m tricarbonyl-HBED-CC complexes and structural prediction in solution by density functional theory

Manuscript ID: RSOS-191247.R1

It is a pleasure to accept your manuscript in its current form for publication in Royal Society Open Science. The chemistry content of Royal Society Open Science is published in collaboration with the Royal Society of Chemistry.

The comments of the reviewer(s) who reviewed your manuscript are included at the end of this email. I apologise this has taken longer than usual.

RSC Associate Editor:
Comments to the Author:
(There are no comments.)

RSC Subject Editor:
Comments to the Author:
(There are no comments.)

Reviewer(s)' Comments to Author:
Reviewer: 2

Comments to the Author(s)
Comments to the Author

I appreciate the effort of the revised manuscript presented following the suggestions from reviewers.

The manuscript has been revised well. I think this manuscript will be acceptable.

Appendix A

Dear Editors and Reviewers:

Thank you very much for your letter and for the reviewers' comments concerning our manuscript(RSOS-191247), entitled "Synthesis of novel technetium-99m tricarbonyl-HBED-CC complexes and structural prediction in solution by density functional theory". Those comments are all valuable and very helpful for revising and improving our paper and they are also of great guiding significance to our researches. We have carefully gone through these comments and made the corresponding corrections. Revised portions in the manuscript are marked in red for recognizing.

We have made corresponding adjustments to each reviewer's comments, as detailed below:

Reviewer: 1

1. The standard B3LYP functional was used for the calculations. The only concern is that dispersion was not included, and single point calculations with Grimme's dispersion correction should be added.

Response:

We added the calculations about the single point enthalpies and energies calculations with dispersion interaction effects were proposed as B3LYP-D3 with SMD solvent model. The stability order of **a1-a5** and **b1-b3** is still the same as that without dispersion correction, the corresponding results have been stated in text(in red).

2. It seems that electronic energies or enthalpies were reported, but Gibbs energies may be reported for comparison.

Response:

We reported the Gibbs free energies with thermal correction calculated by B3LYP+IDSCRF/DGDZVP method, not electronic energies. We have not stated clearly before, but now we have noted in manuscript.

3. Solvation effect in water (if in aqueous solution) should be considered using the SMD solvation model.

Response:

The solvent (water) effect was considered by the polarized continuum model (PCM), I am very sorry that there is no clear explanation in the previous version. We have performed SMD solvation calculation, and the results are given in Supporting information.

4. Coordinates should be given in the supporting information.

Response:

We have put coordinates and energies in the supporting information for present version.

Reviewer: 2

For example, the IR spectrum in the electronic supplementary material is upside down !

Response:

I am really sorry for making such a low-level mistake. The IR spectrum in the electronic supplementary material was fixed.

In addition, in the HRMS spectrum (Fig2 in ESM) of $[\text{Re}(\text{CO})_3\text{L}_2]$, there is no data corresponding to the measured values described in the experimental section (HRMS calcd. For $\text{ReC}_{39}\text{H}_{43}\text{N}_{10}\text{O}_{15} (\text{M}+\text{H})^+$: 1079.7, found 1079.7827). Please check other experimental data described in the ESM.

Response:

I have fixed the HRMS data described in the experimental section. Thank you so much for the careful review by the reviewer. As shown in line 210. Other experimental data described in the ESM was checked carefully.

For the DFT calculation results, the coordinates of the calculation results for each structure should be described in ESM. If authors had performed the calculations in the solution, the authors should refer to the details of how to handle calculations in solution (for example, the polarizable continuum model (PCM) to include solvent effects).

Response:

We have done that.

After receiving the comments of the reviewer, I carefully checked the manuscript. Many errors and incorrect descriptions were found and revised. In addition to the several issues mentioned by reviewers, I also fixed the Fig 3, the title of Fig 6, the format of some references (1, 35, 36, 43) and some syntax errors. For example line 340 change “**a2, a3 and a4**” to “**a2, a3 and a4**”. “space” was added when it was needed, for example in line 207-209. Line 272, “chelators” was changed to “chelator”. Line 288, “figure” was changed to “figures” and “was” was changed to “were”. Line 302, “ especially ” was added. Line 308, “3E” was changed to “5E” Line 314, “chromatograms” was changed to “chromatogram”. Line 319, “structure” was changed to “structures” and “complex” was changed to “complexes”. Line 368, “are” was changed to “were”. Line 392, “examples” was changed to “example” and “report” was changed to “reported”. Line 401, “predicts” was changed to “predicted”. Line 405, “is” was changed to “was”. Line 406, “chromatogram” was changed to “chromatograms”.

Reply: Thank you very much for taking care of our manuscript, we have changed accordingly.

Best regards,

Fang De-cai and Zhu Lin